# Characteristics and Residual Health Risk of Organochlorine Pesticides in Fresh Vegetables in the Suburb of Changchun, Northeast China

**DOI:** 10.3390/ijerph191912547

**Published:** 2022-10-01

**Authors:** Nan Wang, Zhengwu Cui, Yang Wang, Jingjing Zhang

**Affiliations:** 1Northeast Institute of Geography and Agroecology, Chinese Academy of Sciences, Changchun 130102, China; 2University of Chinese Academy of Sciences, Beijing 100049, China; 3College of Resources and Environment, Henan Agricultural University, Zhengzhou 450002, China

**Keywords:** suburban vegetables, organochlorine pesticides, maximum residue limit (MRL), sources identification, target hazard quotient (THQ)

## Abstract

In this study, eleven organochlorine pesticides (OCPs) in fresh vegetables in the Changchun suburb were investigated, and their potential health risks were evaluated. The average concentrations of OCPs in edible parts of vegetables were found in the following descending order: Σhexachlorocyclohexanes (ΣHCHs) (6.60 µg·kg^−1^) > Σdichlorodiphenyltrichloroethanes (ΣDDTs) (5.82 µg·kg^−1^) > ΣChlordanes (2.37 µg·kg^−1^) > heptachlor (0.29 µg·kg^−1^). Moreover, OCPs in different types of vegetables exceeded the maximum residue limits (MRLs), and the exceeding rates in various vegetables decreased in the following order: leafy vegetables (19.12%) > root vegetables (18.75%) > fruit vegetables (3.85%). The proportions of OCPs exceeding MRL in different vegetables were found in the following descending order: Welsh onion (22.50%) > radish (18.75%) > Chinese cabbage (14.29%) > pepper (6.90%) > cucumber (3.23%) > eggplant (2.94%) > tomato (2.78%). The sources’ identification results showed that DDTs in vegetables came mainly from newly imported technical DDTs and dicofol, while HCHs originated mainly from lindane. For both adults and children, the average target hazard quotients (avg. THQ) were all less than 1, and the average hazard index (avg. HI) values were 0.043 and 0.036, respectively. There were no significant health risks associated with OCP exposure for the inhabitants of the study area.

## 1. Introduction

Organochlorine pesticides (OCPs) are a class of typical organic pollutants with high toxicity, persistence, and high bioaccumulation and pose a threat to human health by penetration through the food chain [1]. Numerous studies have revealed that, after entering the human body, OCPs could cause toxic effects on the endocrine, immune, and nervous systems, whereas some OCPs have even exhibited carcinogenic effects [2]. Organochlorine pesticides have played an important role in controlling agricultural pests and diseases and increasing agricultural production and farmers’ income in China. During 1950–1980, China has been one of the countries with the largest production and use of OCPs around the world, with the cumulative application of dichlorodiphenyltrichloroethanes (DDTs) and hexachlorocyclohexanes (HCHs) of about 0.4 × 10^5^ t and 4.9 × 10^5^ t, respectively [3]. Although OCPs have been banned in China’s agricultural system for nearly 40 years, they are still frequently detected in environmental media (such as soil, water, and sediments) and food (such as vegetables, fruits, and fish) because of their strong chemical stability and long-distance migration [3,4].

Vegetables have become one of the major foods in people’s daily diet due to their high nutritional value and provide various vitamins, minerals, dietary fiber, and antioxidants for the human body. The physiological characteristics of vegetables make them more susceptible to pesticide pollution than crops [5]. Vegetables will suffer from extensive pests and diseases during their growth. Therefore, it is necessary to frequently use OCPs to control pests, which also ensures the yield and quality of vegetables. Although the use of OCPs has brought great economic benefits, their residues in vegetables and health risks to consumers have attracted widespread attention. Research on the residues and distribution of OCPs in tomatoes showed that the total residues of OCPs in tomatoes were in the range of 0.062–0.402 µg·kg^−1^, whereas DDTs, heptachlor, and dieldrin was also detected [6]. The maximum (max) residues of OCPs in cucumbers could reach 1.628 µg·kg^−1^ [7]. The monitoring results of OCPs in vegetables which were sold in Beijing (China) markets showed that the residues of DDTs and HCHs were up to 10.4 µg·kg^−1^ and 58.8 µg·kg^−1^, respectively [8]. Liang et al. (2021) found that the max residues of HCHs and heptachlor in edible parts of vegetables in the southern Leizhou Peninsula (China) exceeded the national standard limit value, whereas the max values of the target hazard quotient (THQ) and hazard index (HI) in Chinese chives, and pepper samples were greater than 1, which might be a threat to human health [9]. Although OCPs have been banned for decades, their detection rates and residues in fresh vegetables are still high, due to which their potential health hazards to consumers cannot be ignored.

As a transitional zone between agricultural production and urban living, suburban areas have become important production bases for regional vegetable products. Regarding the transportation costs and preservation requirements of vegetables, more and more suburban farmland is used to cultivate vegetables, and therefore, suburban vegetable production has become an important part of the regional agricultural economy [10]. With the growing demand for vegetables among urban residents, the degree of cultivation intensity of suburban farmland has increased significantly. The excessive use of chemical fertilizers and pesticides and the discharge of domestic and industrial wastes posed serious challenges to the quality of the suburban environment and food safety. At present, most of the research studies on the status of pollution in suburban vegetables mainly focus on heavy metals and polycyclic aromatic hydrocarbons, which come from traffic and industrial emissions and can contaminate vegetables through atmospheric deposition [11,12]. There are only a few studies focusing on pesticide residues in suburban vegetables [13], especially the OCPs. Therefore, it is necessary to conduct research on the residues and risks of OCPs in vegetables in suburban areas.

Located in northeastern China, Changchun is an important agricultural production base. Vegetables constitute a major part of the diet of local inhabitants. Natives have some special dietary habits for vegetables, such as direct consumption of raw vegetables (such as Welsh onions, radishes, cucumbers, and tomatoes), and also store some non-perishable vegetables in autumn for consumption during winter–spring. Vegetables are of great significance to local residents, and it is important to understand the edible safety of vegetables in the region. In view of this, seven kinds of vegetables were collected in the suburbs of Changchun with the purposes of (1) quantifying the concentrations of OCPs in fresh vegetables; (2) identifying the potential sources of OCPs; (3) calculating the THQ and HI of OCPs to evaluate the health risks of consuming the vegetables. The results will provide evidence for policymakers to take targeted measures to reduce potential health risks from intoxicated vegetables and ensure the safe consumption of suburban vegetables.

## 2. Materials and Methods

### 2.1. Study Area and Sampling Sites

Changchun, the capital of Jilin Province, is located in northeast China (43°43′ N, 125°19′ E) and is also an important crop and commodity grain base in China. Changchun belongs to the continental monsoon climate with an annual average temperature of 4.8 °C and annual average precipitation of 569.6 mm. The main soil type is typical phaeozem. The vegetable-cultivated area in the suburbs of Changchun is about 3617 hm^2^, whereas the vegetable yield is about 9.8 × 10^4^ tons.

The suburbs of Changchun were divided into a grid of 1 × 1 km cells, and 54 typical vegetable plots were selected (Figure 1). A handheld global positioning system (GPS) was used to record all sampling points. During September 2018–October 2018, 214 vegetable samples of seven types were collected and mainly included Chinese cabbage, Welsh onion, cucumber, pepper, eggplant, tomato, and radish. These vegetables are commonly planted in the local area. The same types of vegetables were collected in triplicates as sub-samples from each site and mixed thoroughly to obtain a representative sample. Approximately 500 g of the vegetable samples were packed in a polyethylene zip-lock bag and, after being numbered, transported to the laboratory. Vegetable samples were washed with deionized water and stored in a −20 °C refrigerator.

### 2.2. Materials and Reagents

OCPs standard solutions were purchased from the National Sharing Platform for Reference Materials in China, including eight OCPs mixed standard solutions (α-HCH, β-HCH, γ-HCH, δ-HCH, o, p′-DDT, p, p′-DDT, p, p′-DDD, and p, p′-DDE) and single standard solutions (cis-chlordane, tran-chlordane, and heptachlor). Acetone and n-hexane (high-performance liquid chromatography grade), anhydrous sodium sulfate (analytical grade), activated alumina (40–60 mesh), and silica gel (60–100 mesh) were obtained from Aladdin Chemical Corp., China. Anhydrous sodium sulfate was dried for 3 h at 400 °C before use and after cooling, stored in a desiccator for subsequent use. The silica gel was activated for 12 h at 180 °C. The filter paper and cotton thread used to wrap the samples in Soxhlet extraction were sequentially washed with acetone, n-hexane, and distilled water in an ultrasonic cleaner and dried in a baking oven.

### 2.3. Analysis and Quality Control

The Soxhlet extraction and purification process of OCPs followed the previously reported modified method [14].

Extraction: Vegetable samples were cut into small pieces. First, 30.0 g of samples were accurately weighed, and an appropriate amount of anhydrous sodium sulfate was added to them. After grinding in the mortar, the mixtures were put into the Soxhlet extractor, and a 100 mL n-hexane-acetone mixture (1:1, *v*:*v*) was added to extract for 24 h. The extracts were concentrated using a rotary evaporator to nearly dry, and 5 mL n-hexane was added to redissolve to complete the solvent conversion.

Purification: First, 1 g anhydrous sodium sulfate, 4 g active silica gel, 2 g active alumina, and 1 g anhydrous sodium sulfate were successively filled into the glass column from bottom to top. The extract was transferred to a purification column that was pre-activated with 40 mL n-hexane. The effluent was discarded. Subsequently, the column was eluted with 30 mL mixed liquor of acetone-n-hexane (1:9, v:v), and the eluate was collected. The eluent was concentrated with a rotary evaporator and dissolved in 1 mL of n-hexane to determine its concentration.

Analysis: The OCPs were analyzed using gas chromatography (GC-2010, Shimadzu, Japan) with the electron capture detector (ECD), equipped with an HP-5 chromatographic column (30 m × 0.25 mm × 0.25 μm film thickness, USA). High-purity nitrogen (purity > 99.99%) was used as carrier gas. The injection and detector temperatures were set to 260 °C and 300 °C, respectively. One microliter of the extract was injected in the splitless mode. The temperature programming of GC was set as follows: the initial temperature at 100 °C for 1 min, ramping to 190 °C at 12 °C/min, and held for 8 min, followed by continuous ramping to 250 °C at 3 °C/min, and held for 10 min. The characteristic chromatogram of 11 OCPs is presented in Figure 2.

The analytical procedures used in this study were conducted under strict quality assurance and quality control. Quality assurance and control were conducted using procedural blanks, spiked blanks, and duplicate samples. A signal-to-noise ratio of 3 was used to calculate the limits of detection (LODs). Recoveries were determined by spiking the blank vegetable samples with a standard mix solution, and the spiked concentration was set to 50.00 µg·kg^−1^. Each sample was spiked in triplicate and then analyzed using the proposed method. Table 1 presents the LODs, spike concentration, recoveries, and relative standard deviations (RSDs). The compounds to be tested in the blank samples were below the LODs and would not affect the determination of the actual sample. For quantitative OCP, concentrations below the LODs were considered non-detectable and set as zero in the calculations. A standard of 0.10, 0.25, 0.50, 1.00, 2.00, and 4.00 μg·mL^−1^ was used to quantify the calibration curves, and the linear relationship of R^2^ > 0.99 was obtained. All concentrations of OCPs were based on a fresh basis.

### 2.4. Human Risk Assessment of OCPs through Vegetables Consumption

The target hazard quotient (THQ) was applied to evaluate the human health risk of OCP residues in the vegetables [15]. Due to the differences between adults and children in the daily intake of vegetables and the tolerance limits of OCPs, the health risks of adults and children were evaluated separately.

The estimated daily intake (EDI, µg·kg^−1^·d^−1^) depends on both the individual OCP concentrations and the daily consumption of food [13]. The EDI is calculated using Equation (1).
(1)EDI=C×Con/BW
where *C* is the concentration of OCPs in vegetables (µg·kg^−1^), *Con* is the average daily vegetable consumption of local inhabitants (242.00 g·d^−1^ for adults and 108.50 g·d^−1^ for children) [16], and *B**W* (kg) represents the average body weight (55.90 kg for adults and 32.70 kg for children) [13].

As the evaluation standard, the THQ was based on the ratio of EDI to acceptable daily intake (ADI). The ADI was obtained based upon the standard GB 2763-2021, China (National food safety standard—Maximum residue limits for pesticides in food) [17] and listed in Appendix A. The THQ is calculated using Equation (2).
(2)THQ=EDI/ADI

The multiple health risks of various OCPs in vegetables were represented by the hazard index (HI). Moreover, based upon the daily average consumption of vegetables for a human being, HI is calculated using Equation (3).
(3)$$\mathrm{HI}=\sum_{n=1}^{i}THQ_{n}$$

If the value of THQ or HI was less than 1, there was no obvious health risk. However, if the value of THQ or HI was greater than 1, there was a possibility of obvious toxic effects on human health. Furthermore, with the increase in the value of THQ, the probability of toxic impacts increased.

## 3. Results and Discussion

### 3.1. OCPs in Vegetables

The overall detection rate of 11 OCPs in the collected vegetable samples was 55.61%, with the highest detection rate of 70.59% in leafy vegetables, followed by root vegetables (62.50%) and fruit vegetables (46.92%) (Appendix A). The detection rates of OCPs in different vegetables were found to be in the following descending order: Welsh onion (77.50%) > radish (62.50%) > Chinese vegetables (60.71%) > cucumber (51.61%) > pepper (48.28%) > tomato (47.22%) > eggplant (41.18%). The detection rates of different OCPs in vegetables decreased in the following order: ΣHCHs (30.84%) > ΣDDTs (27.57%) > ΣChlordane (19.62%) > heptachlor (13.08%). Although HCHs and DDTs have been banned in China for decades, they still have a relatively high detection rate in suburban vegetables.

The statistical characteristics of the residual concentrations of OCPs in different vegetables are presented in Table 2. The average concentrations of OCPs were in the following descending order: ΣHCHs (6.60 µg·kg^−1^) > ΣDDTs (5.82 µg·kg^−1^) > ΣChlordanes (2.37 µg·kg^−1^) > heptachlor (0.29 µg·kg^−1^). The average concentrations of HCHs and DDTs were 9.43–11.94 µg·kg^−1^ and 9.34–11.43 µg·kg^−1^ in leafy vegetables, 8.42 µg·kg^−1^ and 8.36 µg·kg^−1^ in root vegetables, and 1.91–6.64 µg·kg^−1^ and 1.34–5.64 µg·kg^−1^ in fruit vegetables, respectively. Compared with root and fruit vegetables, the OCPs in leafy vegetables showed higher concentrations, which was consistent with the results of Chourasiya et al. (2015) [18]. The larger leaf surface area of leafy vegetables makes them more susceptible to exposure to OCP pesticides through dry and wet depositions under atmospheric conditions [19]. Some studies reported that the absorption capacity of leafy vegetables to OCPs was higher than that of root vegetables [20], and the higher humidity in fruit vegetables would promote the degradation of pesticides [13,21]. Welsh onion (11.94 µg·kg^−1^) had the highest average concentration of HCHs among all vegetables, followed by Chinese cabbage (9.43 µg·kg^−1^) and radish (8.42 µg·kg^−1^). Vegetables with lower HCH residues were eggplant (3.14 µg·kg^−1^) and cucumber (1.91 µg·kg^−1^). The concentrations of DDTs in different vegetables showed the same concentration characteristic of: Welsh onion > Chinese cabbage > radish > pepper > tomato > eggplant > cucumber.

In order to strengthen the control of pesticide residues in food, China implemented the national food safety standard—maximum residue limits of pesticides in food (GB 2763-2012, China) in 2012. The standard stipulated the maximum residue limits (MRLs) of various pesticides in different kinds of foods and supplemented and amended the standard in 2014, 2019, and 2021, respectively. The proportions of 11 OCPs exceeding the corresponding MRLs in edible parts of different vegetables are listed in Appendix A. The residual concentrations of OCPs in 9.81% of the samples exceeded the MRLs, whereas the exceeding rates of OCPs in different vegetables followed the descending order of: Welsh onion (22.50%) > radish (18.75%) > Chinese cabbage (14.29%) > pepper (6.90%) > cucumber (3.23%) > eggplant (2.94%) > tomato (2.78%). The proportion of OCPs’ concentration exceeding MRLs was highest in the leafy vegetables (19.12%), followed by root vegetables (18.75%) and fruit vegetables (3.85%). The exceeding rates of ΣHCHs, ΣDDTs, and ΣChlordane in vegetables were 2.80%, 3.74%, and 4.21%, respectively, whereas heptachlor did not exceed the standard. The exceeding rates of ΣHCHs, ΣDDTs, and ΣChlordanes in different vegetables were found to be in the descending order of: Welsh onion (7.50%) > radish (6.25%) > Chinese cabbages (3.57%) > pepper (3.44%) > cucumber = eggplant = tomato (0%), Welsh onion (10.00%) > Chinese cabbage (7.14%) > radish (6.25%) > pepper (3.44%) > cucumber = eggplant = tomato (0%), and Welsh onion (7.50%) > Chinese cabbages (7.14%) > radish (6.25%) > cucumber (3.23%) > eggplant = tomato (2.77%) > pepper (0%), respectively. The proportion of ΣChlordane exceeding MRLs in vegetables was higher than those of HCHs and DDTs. The main components in technical chlordane were cis-chlordane (13%), tran-chlordane (11%), and heptachlor (5%). Currently, technical chlordane has been widely used as termiticide for buildings, dams, and cable wires [22]. Moreover, it has also been used in green spaces to control termites in recent years [23]. Changchun is one of the famous garden cities in China, with a total green area of 180 km^2^ and a greening rate of 36.5% [24]. Because of its special geographical location, suburban vegetable fields are adjacent to or surrounded by urban green spaces. Therefore, chlordane applied in the urban green space will directly or indirectly enter the suburban vegetable fields, resulting in high chlordane concentrations in vegetables and high proportions of exceeding MRLs [25,26].

### 3.2. Composition and Source Analysis of OCPs

The residues of HCHs and DDTs in suburban vegetables of Changchun were compared with those in other regions around the world, and it was found that the average concentration of DDTs in the vegetables from Changchun suburbs (5.82 µg·kg^−1^) was significantly higher than those in Taizhou, China (0.30 µg·kg^−1^) [27], city of Northwest Russian (0.11 µg·kg^−1^) [28], Taiwan (2.51 µg·kg^−1^) [29] and Cambodia (1.85 µg·kg−1) [30], and Delhi, India (4.53 µg·kg^−1^) [18]. However, the concentration of DDTs in the vegetables from Changchun suburbs was significantly lower than that in Cape Town, South Africa (53.65 µg·kg^−1^) [20]. The average residue of HCHs (6.6 µg·kg^−1^) was much higher than those of the city of Northwest Russia (0.07 µg·kg^−1^) [28], Cambodia (0.47 µg·kg^−1^) [30] and Taiwan (3.78 µg·kg^−1^) [29], whereas it was lower than that of Delhi, India (76.55 µg·kg^−1^) [18].

The main components of DDTs in Changchun suburban vegetables were o, p′-DDT and p, p′-DDT, which accounted for 47.08% and 26.98% of the ΣDDTs, respectively (Figure 3). Technical DDT mixtures mainly consist of p, p′-DDT (75%), o, p′-DDT (15%), p, p′-DDE (5%), and several other trace metabolites [1,31]. Generally, p, p′-DDT can be dechlorinated into p, p′-DDE under aerobic conditions and reduced to p, p′-DDD under anaerobic conditions [32]. The high concentration of p, p′-DDT in vegetables might be due to the presence of various DDT isomers that were not easily degraded in dicofol. Dicofol is one of the most commonly used OCPs in modern agriculture and animal husbandry [20,33]. The ratios of different isomers of OCPs can be used to identify the environmental input information of pesticides. Therefore, the ratio of p, p′-DDT to its degradants can reflect the “new input or historical use” of DDTs in the environment and the residual time. A ratio of W_p, p′-DDT_/W_p, p′-DDE_ of less than 1 indicates that DDTs are from historical inputs, while a ratio of W_p, p′-DDT_/W_p, p′-DDE_ of greater than 1 indicates the new application of DDTs. In this study, the average value of W_p, p′-DDT_/W_p, p′-DDE_ was 2.29 (Figure 4). Except for eggplant, the average ratios of W_p, p′-DDT_/W_p, p′-DDE_ in other vegetables were all higher than 1, indicating that there may be new DDTs input in the study area. In addition, the ratio of o,p′-DDT and p,p′-DDT was about 0.2 in technical DDTs, while the ratio is about 7 in dicofol [1,33]. The ratios of W_o, p′-DDT_/W_p, p′-DDT_ in Changchun suburban vegetables were within the range of 0.46–4.10, which were higher than that of technical DDTs but lower than that of dicofol. It can be inferred that there might be new inputs of technical DDTs and dicofol in the study area, whereas dicofol was the main source.

γ-HCH and β-HCH were the main components of HCHs in suburban vegetables of Changchun, with a cumulative contribution rate of 91.52%, while the contribution rates of α-HCH and δ-HCH were only 6.36% and 2.12%, respectively (Figure 3). Yi et al. (2013) [34] found that the enrichment ability of different isomers of HCHs in the soil-plant system was in the descending order of α- > β- > δ- > γ-HCH. In the present study, the residual of γ-HCH was higher in vegetables, which might be attributed to long-term exposure to lindane-containing pesticides [34]. There are currently two forms of HCHs: technical HCHs and lindane. The technical HCHs are mainly composed of α-HCH (60%~70%), β-HCH (5%~12%), γ-HCH (10%~12%), and δ-HCH (6%~10%), while lindane consists of more than 99% γ-HCH [35]. Compared with α-HCH, γ-HCH is easier to degrade and transform, and its residual time in the environment is shorter. Therefore, the ratio of α-HCH/γ-HCH can be used to monitor the sources of HCHs. When W_α-HCH_/W_γ-HCH_ < 1, it is mainly from the input of lindane, whereas when 3 < W_α-HCH_/W_γ-HCH_ < 7, it is mainly from the input of technical HCHs. Moreover, when W_α-HCH_/W_γ-HCH_ > 7 or 1 < W_α-HCH_/W_γ-HCH_ < 3, it is derived from the historical use of lindane and has degraded to a certain extent. In addition, β-HCH is one of the most stable isomers of HCHs, which is not easy to degrade in the environment. Therefore, β-HCH is usually the most abundant isomer in the environment [32]. The ratio of W_β-HCH_/W_(α + γ)-HCH_ can be used to indicate whether the sources of HCHs are historical residues or new inputs. The ratio of W_β-HCH_/W_(α + γ)-HCH_ > 0.5 indicates a source of historical pollution [18]. According to Figure 4, the ratios of W_α-HCH_/W_γ-HCH_ in all vegetables were less than 1, indicating that there were lindane inputs in the vegetable growing environment. The ratios of W_β-HCH_/W_(α + γ)-HCH_ were within the range of 0.06–1.13, whereas the ratios of leafy vegetables (Chinese cabbage 0.21 and Welsh onion 0.38) were less than 0.5, indicating that HCHs in these two vegetables were mainly derived from the new application of lindane, while the wet and dry deposition of the atmosphere might be the main input route. The ratios of eggplant, pepper, cucumber, and radish were all greater than 0.5. It can be concluded that the HCHs in these four vegetables mainly came from the historical residues of lindane in the soil.

### 3.3. Consumption Health Risk of OCPs in Vegetable

The calculated EDI, THQ, and HI values of different OCPs for children and adults in Changchun are presented in Table 3. The average (avg.) EDI for adults and children were as follows: HCHs > DDTs > chlordane > heptachlor, while max EDI for adults and children were in the decreasing order of DDTs > HCHs > Chlordane > heptachlor. The values of avg. EDI and max EDI for adults were higher than those for children. The max EDI and avg. EDI values of OCPs were lower than the ADI values in national food safety standards in China, indicating that the current vegetable consumption posed a low health risk to the local inhabitants. However, it should be noted that the dietary structure of the population is diverse and complex, and some people may consume more vegetables than the average, such as vegetarians and people undergoing a fat-reducing period. Their EDI will be correspondingly higher. In addition, the ways of vegetable cleaning (detergent or not), eating (raw or cooked), and cooking (boiled or fried) will bring some uncertainties, leading to an overestimation or underestimation of dietary OCPs exposure [13]. Compared with other regions around the world (Table 4), the intakes of HCHs through the consumption of suburban vegetables of inhabitants in Changchun (0.022–0.029 µg·kg^−1^·d^−1^) were higher than those of most other regions except for Taizhou, China (0.137 µg·kg^−1^·d^−1^). In contrast, the intakes of DDTs (0.019–0.025 µg·kg^−1·^d^−1^) of the inhabitants in Changchun were higher than those of Dalian, China (0.003 µg·kg^−1^·d^−1^), Denmark (0.0037 µg·kg−1·d^−1^), and Punjab Province, Pakistan (0.019 µg·kg^−1^·d^−1^), but lower than those of Jakarta, Bogor, and Yogyakarta, Indonesia (0.040 µg·kg^−1^·d^−1^) and Taizhou, China (0.076 µg·kg^−1^·d^−1^). In general, the consumption of OCPs through suburban vegetables by residents in the study area was at a high level.

The max THQs of HCHs, DDTs, Chlordane, and heptachlor for adults were 0.060, 0.038, 0.32, and 0.38, respectively, and those for children were 0.046, 0.029, 0.25, and 0.29, respectively (Table 3). For both adults and children, the max THQ of OCPs exhibited a descending order of heptachlor > Chlordane > HCHs > DDTs. The calculated max HI values for adults and children were 0.80 and 0.62, respectively. Heptachlor, Chlordane, HCHs, and DDTs contributed 47.70%, 40.16%, 7.46%, and 4.68% to the max HI for adults and children, respectively. The avg. HI values of adults and children were 0.043 and 0.036, respectively, and with the descending order of Chlordane > heptachlor > HCHs > DDTs. The contribution rates of HCHs, DDTs, chlordane, and heptachlor to avg. HI were 13.61%, 5.98%, 49.48%, and 30.93%, respectively. Therefore, Chlordane and heptachlor were the key compounds that led to potential health risks for residents. The results suggested that residents may not suffer significant adverse health effects based on the average level of OCPs.

Although daily intake of OCPs through vegetables is an important route of dietary exposure for people, many studies have reported that humans can also be exposed to OCPs through other foods such as rice, vegetable oils, fruits, aquatic products, and meats [5,15,38,41,42]. Wang et al. (2021) [43] investigated the intake of OCPs through diet and related health risks among women of childbearing age in agricultural areas of northern China. Among the seven types of foods investigated, the intakes of OCPs through vegetables and fruits were significantly higher than those of other foods and accounted for 35.1% and 45.6% of the total intake of OCPs, respectively. Hao et al. (2014) [44] studied the DDTs’ residual levels in edible fish in Guangzhou and the consumption risk and found that the daily exposure of inhabitants to DDTs from fish consumption was 23.0–1875.6 ng·d^−1^, whereas the long-term consumption of croaker might have certain potential health risks. A study from Pakistan showed that OCPs’ residuals in cereals were up to 15.2 µg·kg^−1^ (dry weight), whereas the consumption of contaminated cereal crops could pose a health risk to the population in the study area [41]. Furthermore, OCPs ingested by humans through consuming individual types of food may not cause health risks, but there would be a certain possibility of toxic effects on humans under exposure to multiple foods.

In addition, when assessing the health risks of OCPs, special groups such as the elderly, pregnant women, and infants should be considered. Wang (2021) [45] found that the accumulation of OCPs such as β-HCH, γ-HCH, o, p′-DDD in the elderly showed a significant positive correlation with age, whereas the accumulation level gradually increased with age. In addition, the levels of β-HCH and o, p′-DDD in the blood of the elderly were positively associated with the risk of hypertension, hyperlipidemia, and diabetes, and dietary pathways accounted for more than 90% of the total exposure to OCPs. Furthermore, HCHs and DDTs are the most frequently detected POPs in breast milk [46,47]. Additionally, β-HCH is a risk factor for neonatal weight loss and is significantly associated with growth and development indicators such as neonatal length and head circumference [48]. Gyalpo et al. (2012) [49] studied the concentration levels of DDTs in the human body’s fat during the lifetime of human individuals. The results showed that the total concentration of DDTs in the human body was the highest at the age of 2 (75 μg·g^−1^ lipid), which was 2.5 times that of the 20-year-old. Meanwhile, the concentrations of DDTs in primiparous mothers were higher than those in prolific mothers, which resulted in increased exposure to DDTs in neonates. In addition, DDT in humans was mainly derived from the diet. Therefore, future protection policies regarding exposure to OCPs should consider vulnerable groups such as the elderly, pregnant women, and younger infants, as they have a higher sensitivity to pollutants and may be more vulnerable than adolescents and adults.

## 4. Conclusions

In this study, the characteristics, sources, and health risks of residual OCPs in vegetables in Changchun suburbs (China) were studied. The residues of OCPs in leafy vegetables were found to be higher than those in root and fruit vegetables. The residues of OCPs in 9.81% of the vegetable samples were higher than the MRLs, and the ratio of exceeding the MRLs was found in the descending order of leafy vegetables > root vegetables > fruit vegetables. The ratios of OCPs exceeding the MRLs in different vegetables decreased in the following order: Welsh onion > radish > Chinese cabbage > pepper > cucumber > eggplant > tomato. The source identification results showed that the DDTs in vegetables mainly came from a mixed source of technical DDTs and dicofol, of which dicofol was the predominant source. The HCHs in Chinese cabbage and Welsh onion mainly originated from new inputs of lindane, while the HCHs in eggplant, pepper, cucumber, and radish came from the historical residue of lindane in soil. The THQ values of OCPs were all less than 1, whereas the avg. HI values of adults and children were 0.043 and 0.036, respectively. The health risks of OCPs in vegetables were not obvious to the consumers. Although OCPs have been banned in China for decades, they still have high detection rates and residues in vegetables. Therefore, regular inspection of agricultural products and supervision of production, sales, and use of pesticides should be strengthened.

## Figures and Tables

**Figure 1 ijerph-19-12547-f001:**
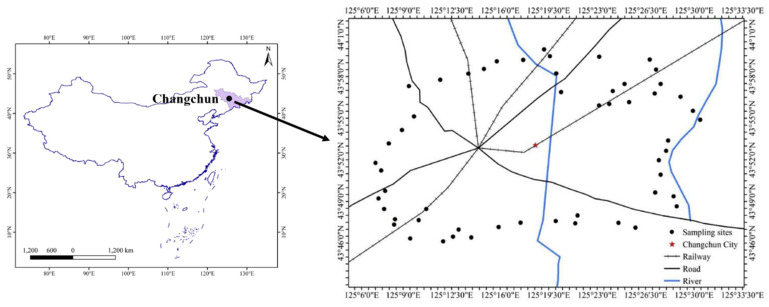
Location of the sampling sites.

**Figure 2 ijerph-19-12547-f002:**
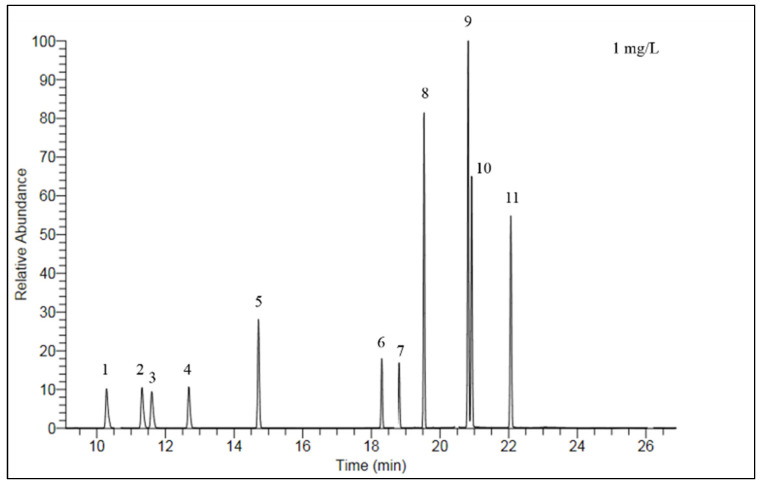
Characteristic chromatogram of 11 OCPs (1. α-HCH; 2. β-HCH; 3. γ-HCH; 4. δ-HCH; 5. heptachlor; 6. cis-chlordane; 7. tran-chlordane; 8. p,p′-DDE; 9. p,p′-DDD; 10. o,p′-DDT; 11. p,p′-DDT).

**Figure 3 ijerph-19-12547-f003:**
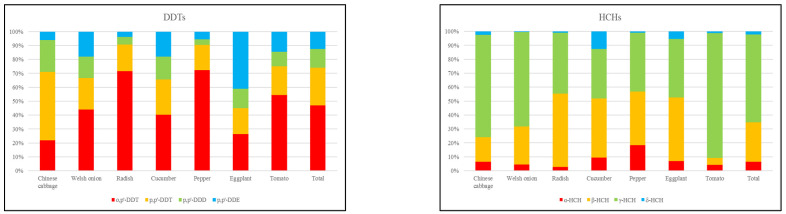
Metabolite composition characteristics of DDTs and HCHs in different vegetables.

**Figure 4 ijerph-19-12547-f004:**
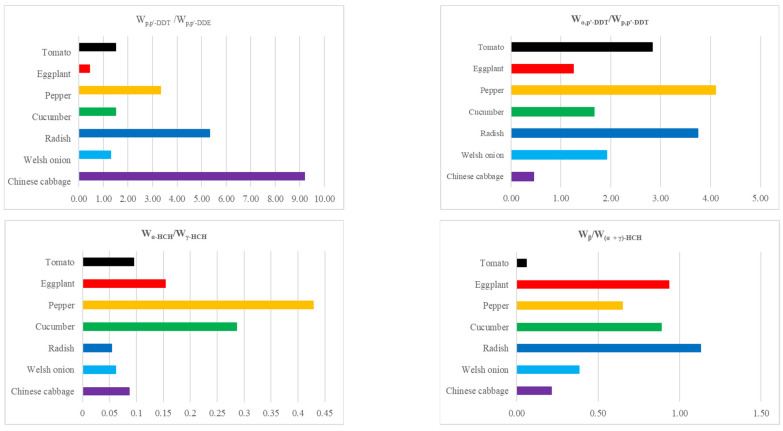
W_p,p′-DDT_/W_p,p′-DDE_, W_o,p′-DDT_/W_p,p′-DDT_, W_α-HCH_/W_γ-HCH_, and W_β/(α + γ)-HCH_ in different kinds of vegetables.

**Table 1 ijerph-19-12547-t001:** The LODs, spiked recoveries, and RSD of the method.

Pesticides	LOD (µg·kg^−1^)	Spiked Concentration (µg·kg^−1^)	Recovery (%) (*n* = 3)	RSD%
o,p′-DDT	0.13	50.00	87.61	5.65
p,p′-DDT	0.17	50.00	83.53	7.32
p,p′-DDD	0.11	50.00	92.95	4.76
p,p′-DDE	0.10	50.00	89.66	9.82
α-HCH	0.080	50.00	91.23	6.45
β-HCH	0.18	50.00	84.93	8.71
γ-HCH	0.090	50.00	93.51	10.20
δ-HCH	0.10	50.00	81.39	9.80
heptachlor	0.11	50.00	76.21	11.15
cis-chlordane	0.070	50.00	80.57	5.81
tran-chlordane	0.080	50.00	81.60	9.14

**Table 2 ijerph-19-12547-t002:** Concentrations of OCPs in edible parts of vegetables from Changchun, China (µg·kg^−1^).

OCPs	Leafy Vegetables (*n* = 68)				Root Vegetables (*n* = 16)		Fruit Vegetables (*n* = 130)								Total (*n* = 214)	
	Chinese Cabbage (*n* = 28)		Welsh Onion (*n* = 40)		Radish (*n* = 16)		Cucumber (*n* = 31)		Pepper (*n* = 29)		Eggplant (*n* = 34)		Tomato (*n* = 36)			
	Range	Mean	Range	Mean	Range	Mean	Range	Mean	Range	Mean	Range	Mean	Range	Mean	Range	Mean
∑HCHs	nd-69.13	9.43	nd-53.76	11.94	nd-53.71	8.42	nd-9.08	1.91	nd-68.57	6.64	nd-26.56	3.14	nd-54.41	5.40	nd-69.13	6.60
α-HCH	nd-5.55	0.62	nd-3.40	0.52	nd-1.68	0.22	nd-2.28	0.18	nd-14.64	1.22	nd-3.04	0.22	nd-1.98	0.23	nd-14.64	0.42
β-HCH	nd-10.02	1.64	nd-14.12	3.28	nd-33.92	4.44	nd-8.61	0.81	nd-20.02	2.56	nd-21.68	1.43	nd-3.90	0.26	nd-33.92	1.88
γ-HCH	nd-67.59	6.93	nd-50.01	8.08	nd-39.94	3.68	nd-5.59	0.68	nd-34.03	2.80	nd-19.96	1.32	nd-47.76	4.84	nd-67.59	4.16
δ-HCH	nd-3.03	0.24	nd-1.58	0.06	nd-0.79	0.08	nd-1.73	0.24	nd-1.08	0.06	nd-2.08	0.17	nd-0.68	0.070	nd-3.03	0.14
∑DDTs	nd-59.65	9.34	nd-56.4	11.43	nd-56.28	8.36	nd-9.29	1.34	nd-86.65	5.64	nd-14.84	2.04	nd-55.99	3.05	nd-86.65	5.82
o,p′-DDT	nd-28.27	2.05	nd-32.77	5.04	nd-37.96	5.98	nd-3.77	0.54	nd-73.19	4.08	nd-5.53	0.54	nd-40.33	1.66	nd-73.19	2.74
p,p′-DDT	nd-31.72	4.60	nd-14.98	2.58	nd-14.46	1.60	nd-4.46	0.34	nd-11.93	1.02	nd-4.21	0.38	nd-6.88	0.63	nd-31.72	1.57
p,p′-DDD	nd-21.54	2.12	nd-9.93	1.76	nd-2.98	0.46	nd-2.88	0.22	nd-2.57	0.23	nd-2.62	0.28	nd-4.59	0.32	nd-21.54	0.78
p,p′-DDE	nd-3.77	0.57	nd-15.42	2.05	nd-2.46	0.32	nd-2.69	0.24	nd-4.62	0.31	nd-7.38	0.84	nd-4.14	0.44	nd-15.42	0.73
∑Chlordans	nd-37.16	2.93	nd-29.55	3.31	nd-21.81	2.74	nd-34.47	1.85	nd-33.74	2.93	nd-29.96	2.02	nd-20.92	1.02	nd-37.16	2.37
cis-chlordane	nd-18.05	1.51	nd-13.66	1.64	nd-16.47	2.16	nd-26.02	1.27	nd-15.41	1.32	nd-22.33	1.58	nd-12.65	0.72	nd-26.02	1.43
tran-chlordane	nd-19.14	1.42	nd-16.42	1.67	nd-5.33	0.58	nd-8.59	0.58	nd-18.22	1.61	nd-7.70	0.44	nd-8.23	0.30	nd-19.14	0.94
heptachlor	nd-7.34	0.66	nd-8.75	0.48	nd-1.51	0.14	nd-4.34	0.27	nd-4.48	0.32	nd-2.32	0.08	nd-6.49	0.33	nd-8.75	0.29

nd not detected; ΣHCHs = α-HCH + β-HCH + γ-HCH + δ-HCH; ΣDDTs = o,p′-DDT + p,p′-DDT + p,p′-DDD + p,p′-DDE; ΣChlordane = cis-chlordane + tran-chlordane.

**Table 3 ijerph-19-12547-t003:** Estimated daily intake and potential health risk of OPs via vegetables.

	OCPs	ADI µg·kg^−1^·d^−1^	Ave EDI µg·kg^−1^·d^−1^	Ave THQ	Ave HI	Max EDI µg·kg^−1^·d^−1^	Max THQ	Max HI
Children	HCHs	5	0.022	0.0040	0.036	0.23	0.046	0.62
	DDTs	10	0.019	0.0020		0.29	0.029	
	Chlordan	0.5	0.0080	0.016		0.12	0.25	
	heptachlor	0.1	0.0010	0.010		0.029	0.29	
Adults	HCHs	5	0.029	0.0060	0.043	0.30	0.060	0.80
	DDTs	10	0.025	0.0030		0.38	0.038	
	Chlordan	0.5	0.010	0.021		0.16	0.32	
	heptachlor	0.1	0.0012	0.013		0.038	0.38	

**Table 4 ijerph-19-12547-t004:** Comparison of DDTs and HCHs daily intake in food from native and foreign regions (µg·kg^−1^·d^−1^).

Sites	HCHs	DDTs	Reference
Dalian, China	0.001	0.003	[36]
Taizhou, China	0.137	0.076	[37]
Punjab Province, Pakistan	0.0039	0.019	[38]
Jakarta, Bogor, and Yogyakarta, Indonesia	0.002	0.040	[39]
Denmark	0.0022	0.0037	[40]
This study	0.022–0.029	0.019–0.025	

## Data Availability

The datasets used and/or analyzed during the current study are available from the corresponding author upon reasonable request.

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
