# Peer review of "Characteristics and Residual Health Risk of Organochlorine Pesticides in Fresh Vegetables in the Suburb of Changchun, Northeast China"

_ijerph, 2022, doi:10.3390/ijerph191912547_

Round 1
Reviewer 1 Report
Title: Characteristics and residual health risk of organochlorine pesticides in fresh vegetables in the suburb of Changchun, northeast China
The paper is well written. The results are new and important to the area. I recommend that you be accepted for publication with MINOR REVISION.
Potentially the material is interesting for International Journal of Environmental Research and Public Health readers, but some questions have to be addressed before the paper would be acceptable for such publication.
Keywords – Replace keywords with others that are not in the title
Introduction
Lines 72-74 - How do heavy metals and PAHs contaminate vegetables?
Lines 74-75 – “There are only a handful of studies focusing on pesticide residues...”. Strange information. Check in the literature.
Methods
- Make a topic about the statistical analysis used. It was not clear whether comparisons were made and which tests were used.
Results and Discussion
Table S1 – Italicize scientific names
Table 2 - Why in the column total range the values ​​differ from the highest found?
Table 3- Insert units
Lines 254-256 - Insert a figure demonstrating DDT and its metabolites
Lines 256-257 - Why is dicofol released if it has important features related to toxicity?
Lines 278-281 - Insert a figure demonstrating the composition of the HCHs
Author Response
Keywords
Point 1: Replace keywords with others that are not in the title
Response: Thank you very much for your suggestion! The suggestion has been adopted. We have replaced some keywords to avoid duplication with the title. Please see the “keywords” section (Line 26-27).
Introduction
Point 2: Lines 72-74 - How do heavy metals and PAHs contaminate vegetables?
Response: Based on our knowledge and published researches, we believe that heavy metals and PAHs emitted by transportation and industry can contaminate vegetables through atmospheric deposition. Related content has been added in the manuscript. Please see line 72-75.
Point 3: Lines 74-75 – “There are only a handful of studies focusing on pesticide residues...”. Strange information. Check in the literature.
Response: We are very sorry that our expression in the manuscript caused you confusion. What we really want to express is that there are few studies on the characteristics of organochlorine pesticide residues in suburban vegetables, as mentioned in the previous sentence. At present, there have been a certain number of studies on pesticide residues in vegetables, such as vegetables sold in the market and vegetables grown in greenhouses, but there are relatively few studies on vegetables grown in suburban areas. We have revised this content to accurately express our research purpose. Please see Line 76-77.
Methods
Point 4: Make a topic about the statistical analysis used. It was not clear whether comparisons were made and which tests were used.
Response: Thank you very much for your suggestion! The suggestion has been adopted. We have supplemented the “2.5 Statistical analysis” section in the manuscript. Please see Line 187-190.
Results and Discussion
Point 5: Table S1 – Italicize scientific names
Response: Thank you very much for your suggestion! The suggestion has been adopted. We have italicized the scientific names in Table S1.
Point 6: Table 2 - Why in the column total range the values ​​differ from the highest found?
Response: Thank you very much for your reminder. We feel very guilty about not carefully verifying the data in the table due to our carelessness. During data statistics, we used different statistical software, which might lead to inconsistent values in the table. We have checked the data and corrected the data in Table 2. Please see Table 2.
Point 7: Table 3- Insert units
Response: Thank you very much for your suggestion! The suggestion has been adopted. We have inserted the units of ADI and EDI in Table 3. Please see Table 3.
Point 8: Lines 254-256 - Insert a figure demonstrating DDT and its metabolites
Response: Thank you very much for your suggestion! The suggestion has been adopted. We have supplemented the composition characteristic figure of DDTs metabolites in the manuscript. Please see Figure. 3 (Line 285-286).
Point 9: Lines 256-257 - Why is dicofol released if it has important features related to toxicity?
Response: Vegetables are affected by various pests during growth, especially mites. A great number of mite fauna (Arachnida: Acari) have been reported on vegetable crops throughout the world. In China, there are about 40 species of mites that are seriously harmful nationally or locally. Dicofol is a broad-spectrum acaricide, which is effective against adult mites, nymph and mite eggs. Dicofol has good selectivity, does not harm the natural enemy, and has a long residual effect. Therefore, dicofol is still widely used as acaricide in various countries, particularly in the United States and China [1].
[1]. Zargar S, A. Wani T. Exploring the binding mechanism and adverse toxic effects of persistent organic pollutant (dicofol) to human serum albumin: A biophysical, biochemical and computational approach. Chemico-Biological Interactions, 2021, 350:109707.
Point 10: Lines 278-281 - Insert a figure demonstrating the composition of the HCHs
Response: Thank you very much for your suggestion! The suggestion has been adopted. We have supplemented the composition characteristic figure of HCHs metabolites in the manuscript. Please see line Figure. 3 (Line 285-286).

Reviewer 2 Report
The problem of long residue of organochlorine pesticides has been attracting worldwide attention. In this study, eleven organochlorine pesticides (OCPs) in fresh vegetables of Changchun suburb were investigated and their potential health risks were evaluated. The manuscript is well written and the data are clearly expressed. The results are helpful to understand the residue of organochlorine pesticides on vegetables in Northeast China. After correcting the following issues, I thought it could be published on ijerph.
1.L105, “Vegetable samples were washed with deionized water and stored in a -20 °C refrigerator”, Generally, we do not use water to clean pesticide residues in primary agricultural products when we detect them. Please explain why we use water to clean and what impact will it have on the test results?
2.L234-237, “Because of its special geographical location, suburban vegetable fields are adjacent to or surrounded by urban green spaces. Therefore, chlordane applied in the urban green space will directly or indirectly enter the suburban vegetable fields, resulting in high chlordane concentrations in vegetables and high proportions of exceeding MRLs”. Is there any basis for the author to reach this conclusion? If there is, you should quote relevant literature. If not, it should be just a guess.
3.Table 1, What concentration was spiked in the recovery test, and how many concentrations are added? I can't see the description in the manuscript. Please add.
Author Response
Point 1: L105, “Vegetable samples were washed with deionized water and stored in a -20 °C refrigerator”, Generally, we do not use water to clean pesticide residues in primary agricultural products when we detect them. Please explain why we use water to clean and what impact will it have on the test results?
Response: We believe that cleaning vegetable samples with deionized water might has little effect on the test results. The purpose of cleaning vegetable samples with deionized water during sample processing is to remove dust, mud and surface sediment which temporarily attached to the surface of vegetables and might be washed away by rainwater. Those temporarily attached matters haven’t entered into the vegetable tissue yet and can’t cause harm to plants or consumers.
Point 2: L234-237, “Because of its special geographical location, suburban vegetable fields are adjacent to or surrounded by urban green spaces. Therefore, chlordane applied in the urban green space will directly or indirectly enter the suburban vegetable fields, resulting in high chlordane concentrations in vegetables and high proportions of exceeding MRLs”. Is there any basis for the author to reach this conclusion? If there is, you should quote relevant literature. If not, it should be just a guess.
Response: Thanks for your suggestion! We have cited relevant references in the manuscript for the basis of our conclusions. Please see Lin 250 (Reference 25-26).
Point 3: Table 1, What concentration was spiked in the recovery test, and how many concentrations are added? I can't see the description in the manuscript. Please add.
Response: Recoveries were determined by spiking the blank vegetable samples with a standard mix solution, and the spiked concentration was set to 50.00 µg·kg-1. We have added relevant content to the manuscript, please see Table 1 and Line 153-156.

Reviewer 3 Report
Manuscript ijerph-1917984 (Characteristics and residual health risk of organochlorine pesticides in fresh vegetables in the suburb of Changchun, northeast China) investigated eleven OCPs in fresh vegetables of Changchun suburb and evaluated their potential health risks. The results indicated that there was no significant health risk associated with OCPs exposure for the inhabitants of the study area. In my opinion, this manuscript can be considering for publication in International Journal of Environmental Research and Public Health after minor revised.
1. It is recommended to retain two significant digits for residue concentrates, recoveries, percentages and other data.
2. Fresh vegetables are very easy to extract. Why did the authors use Soxhlet extraction to extract as a tedious and inefficient method?
3. In the spiked recovery test, what is the spiked concentration level? And what are the range of standard concentrations for linear curve?
4. The concentration units in this manuscript need to be unified.
5. In Table 3, what is the unit of each data?
6. The authors should give out the characteristic sample chromatogram.
Author Response
Point 1: It is recommended to retain two significant digits for residue concentrates, recoveries, percentages and other data.
Response: Thanks for your suggestion! The suggestion has been adopted. We carefully examined the data in the manuscript and retain the necessary data as two significant digits.
Point 2: Fresh vegetables are very easy to extract. Why did the authors use Soxhlet extraction to extract as a tedious and inefficient method?
Response: Before the batch extraction test, we have read a lot of literature and found that in the currently published studies, Soxhlet extraction is mainly used for the extraction of OCPs in vegetation [1-4]. We have done some preliminary experiments and found that compared with ultrasonic, shaking and other methods, although Soxhlet extraction has lower extraction efficiency and larger solvent consumption, its extraction capacity and stability are higher than the first two methods. Since we are going to measure the content of OCPs in fresh vegetable samples, we need a higher sample volume, and the use of ultrasonic or shaking methods still requires a large amount of extraction solvents and multiple extractions (2-3 times). Based on the published research results, preliminary experimental results and laboratory conditions, we believe that the Soxhlet extraction method is more suitable for the extraction of OCPs from fresh vegetables in this study.
[1]. Mumtaz M, Qadir A, Mahmood A, et al. Human health risk assessment, congener specific analysis and spatial distribution pattern of organochlorine pesticides (OCPs) through rice crop from selected districts of Punjab Province, Pakistan. Science of the Total Environment, 2015, 511:354-361.
[2]. Zhang A, Luo W, Sun J, et al. Distribution and uptake pathways of organochlorine pesticides in greenhouse and conventional vegetables. Science of the Total Environment, 2015, 505:1142-1147.
[3]. Mahmood A, Malik R N, Li J, et al. Human health risk assessment and dietary intake of organochlorine pesticides through air, soil and food crops (wheat and rice) along two tributaries of river Chenab, Pakistan. Food & Chemical Toxicology, 2014, 71:17-25.
[4]. Shen L, Xia B, Dai X. Residues of persistent organic pollutants in frequently-consumed vegetables and assessment of human health risk based on consumption of vegetables in Huizhou, South China. Chemosphere, 2013, 93(10):2254-2263.
Point 3: In the spiked recovery test, what is the spiked concentration level? And what are the range of standard concentrations for linear curve?
Response: Recoveries were determined by spiking the blank vegetable samples with a standard mix solution, and the spiked concentration was set to 50.00 µg·kg-1. A standard of 0.10, 0.25, 0.50, 1.00, 2.00 and 4.00 μg·mL-1 was used to quantify the calibration curves, and the linear relationship of R2 > 0.99 was obtained. We have added relevant content to the manuscript, please see Line 153-156 and Line 159-161.
Point 4: The concentration units in this manuscript need to be unified.
Response: Thanks for your suggestion! The suggestion has been adopted. We have carefully checked the concentration units in the manuscript and made corrections to maintain a consistent format.
Point 5: In Table 3, what is the unit of each data?
Response: The units of ADI and EDI in Table 3 were µg·kg-1·d-1. We have inserted the units of ADI and EDI in Table 3. Please see Table 3.
Point 6: The authors should give out the characteristic sample chromatogram.
Response: Thanks for your suggestion! The suggestion has been adopted. We have added the characteristic chromatogram of 11 OCPs in the “2.3. Analysis and quality control” section. Please see Figure. 2 (Line 147-149).
